# Transcriptomic Dysregulation of Inflammation-Related Genes in Leukocytes of Patients with Gestational Diabetes Mellitus (GDM) during and after Pregnancy: Identifying Potential Biomarkers Relevant to Glycemic Abnormality

**DOI:** 10.3390/ijms232314677

**Published:** 2022-11-24

**Authors:** Andrzej Zieleniak, Monika Zurawska-Klis, Katarzyna Cypryk, Lucyna Wozniak, Marzena Wojcik

**Affiliations:** 1Department of Structural Biology, Faculty of Biomedical Sciences, Medical University of Lodz, 90-752 Lodz, Poland; 2Department of Internal Diseases and Diabetology, Medical University of Lodz, 92-213 Lodz, Poland

**Keywords:** abnormal glucose tolerance (AGT), area under the ROC curve (AUC), cytokines, diagnostic and predictive biomarkers, gestational diabetes mellitus (GDM), inflammation, receiver operating characteristic (ROC), transcriptomic study

## Abstract

Although the immune system has been implicated in the pathophysiology of gestational diabetes mellitus (GDM) and postpartum abnormal glucose tolerance (AGT), little is known about the transcriptional response of inflammation-related genes linked to metabolic phenotypes of GDM women during and after pregnancy, which may be potential diagnostic classifiers for GDM and biomarkers for predicting AGT. To address these questions, gene expression of *IL6*, *IL8*, *IL10*, *IL13*, *IL18*, *TNFA*, and the nuclear factor κB (NFκB)/*RELA* transcription factor were quantified in leukocytes of 28 diabetic women at GDM diagnosis (GDM group) and 1-year postpartum (pGDM group: 10 women with AGT and 18 normoglycemic women), using a nested RT-PCR method. Control pregnancies with normal glucose tolerance (NGT group; *n* = 31) were closely matched for maternal age, gestational age, pre-pregnancy BMI, pregnancy weight, and gestational weight gain. Compared with the NGT group, *IL8* was downregulated in the GDM group, and *IL13* and *RELA* were upregulated in the pGDM group, whereas *IL6*, *IL10*, and *IL18* were upregulated in the GDM and pGDM groups. The *TNFA* level did not change from pregnancy to postpartum. Associations of some cytokines with glycemic measures were detected in pregnancy (*IL6* and *RELA*) and postpartum (*IL10*) (*p* < 0.05). Receiver operating characteristic (ROC) curves showed that *IL6*, *IL8*, and *IL18,* if employed alone, can discriminate GDM patients from NGT individuals at GDM diagnosis, with the area under the ROC curves (AUCs) of 0.844, (95% CI 0.736–0.953), 0.771 (95% CI 0.651–0.890), and 0.714 (95% CI 0.582–0.846), respectively. By the logistic regression method, we also identified a three-gene panel (*IL8*, *IL13*, and *TNFA*) for postpartum AGT prediction. This study demonstrates a different transcriptional response of the studied genes in clinically well-characterized women with GDM at GDM diagnosis and 1-year postpartum, and provides novel transcriptomic biomarkers for future efforts aimed at diagnosing GDM and identifying the high risk of postpartum AGT groups.

## 1. Introduction

Gestational diabetes mellitus (GDM), defined as a carbohydrate intolerance with onset or first recognition during pregnancy, is the most common metabolic disorder during pregnancy, and its prevalence is steadily increasing worldwide, mainly due to a growing number of pregnant women who are overweight/obese and to women who give birth at a more advanced age [1]. GDM affects an estimated 1–14% of pregnant women, depending on ethnicity and the diagnostic criteria applied [2]. In Poland, the prevalence of GDM was reported as 7.49% in 2012 in a study using the diagnostic criteria set by the World Health Organization (WHO)/International Association of Diabetes and Pregnancy Study Groups (IADPSG), with the cutoffs of >90 mg/dL for fasting plasma glucose (FPG) concentration and of ≥140 mg/dL for 2 h plasma glucose concentration at a 75 g oral glucose tolerance test (OGTT) [3]. Of great concern, GDM associates with an increased incidence of complications in both the mother (e.g., gestosis, urinary tract infections, cesarean sections and perinatal injuries) and the fetus (e.g., macrosomia, hypoglycemia, and cardiomyopathy) [4]. Furthermore, GDM constitutes a high risk for future abnormal glucose tolerance (AGT) and cardiovascular disease, including pre-diabetic states (i.e., isolated impaired fasting glucose (IFG), isolated impaired glucose tolerance (IGT), or a combination of IFG and IGT) and type 2 diabetes mellitus (T2DM) [5,6]. For women with GDM, the risk of progression to T2DM has been estimated to be approximately 50% after 10 years [7], with the highest T2DM incidence found in the first 5 years postpartum [8], which, however, may be different depending on the diagnostic criteria used and duration of follow-up [9]. A recent systematic review and meta-analysis revealed that women with prior GDM have a nearly 10-fold higher risk of developing T2DM than those with normal glucose tolerance (NGT) in pregnancy [10]. Of note is that up to 60% of subjects with IGT or IFG will subsequently develop overt diabetes [11]. Therefore, there is an urgent need to identify specific predictive biomarkers related to postpartum AGT among individuals with GDM, which may facilitate the application of appropriate individual lifestyle or/and pharmacological interventions to prevent or delay the progression of GDM to postpartum AGT [12,13]. 

Like T2DM, GDM has a complex pathology that is hallmarked by insulin resistance and inadequate insulin secretion resulting from pancreatic β-cell dysfunction. Although there is no consensus regarding the underlying pathophysiology events during and after diabetic pregnancy, the immune system and inflammatory processes have been recognized as important pathological contributors [14,15]. To date, a long list of GDM-associated mediators of inflammation has been proposed, including numerous cytokines and components of intracellular inflammatory signaling pathways; however, reports of changes in plasma levels of various pro- and anti-inflammatory cytokines during normal pregnancy and GDM are equivocal, in particular, due to differences in patient selection criteria, gestational age at sampling, type of sample, assay methods to measure the concentration of cytokines, and also to the lack of adjustment for maternal body mass index (BMI) [16]. Moreover, some inflammatory mediators circulating in maternal blood in GDM pregnancies are the cumulative secretion of different cells which have major roles in the etiology of GDM [17]. Hence, current research should be shifting to focus on identifying a unique cellular expression pattern of both pro-and anti-inflammatory regulators that are associated with metabolic abnormalities in patients with GDM during and after pregnancy to better understand the existence of an immune–metabolic relationship in these periods. From a clinical perspective, such an approach might also significantly help in finding potential predictive biomarkers for postpartum AGT in women with previous GDM. Keeping this in mind, along with the fact that temporal changes in expression profiles of inflammatory-relevant genes during and after GDM are not well understood, we have undertaken this study to (i) examine changes of leukocyte transcriptional responses in several pro-inflammatory (*TNFA, IL6*, *IL8*, *IL18,* and *RELA* encoding the nuclear factor κappa-B transcription factor p65) and anti-inflammatory (*IL10* and *IL13*) genes in clinically well-characterized GDM patients at the time of GDM diagnosis and 1-year postpartum vs. pregnant women with NGT (these genes were chosen based on previous reports on their importance in GDM patients [16]), (ii) explore possible gene–gene expression correlations, as well as relationships between the expression of the studied genes and clinical phenotypes of the patients during and after diabetic pregnancy, (iii) find gene(s) relevant to immune response and inflammation which the expression could discriminate between the GDM and NGT pregnant women at the time of GDM diagnosis, and, finally, (iv) identify transcriptomic and anthropometric/metabolic predictors of AGT development at 1 year after GDM.

In this study, we used leukocytes as an experimental cellular model because they are not only well-known drivers of the inflammatory process but also are easy to obtain in large volume. Furthermore, these cells may reflect pathological changes elsewhere in the body, as suggested in the literature [18]; therefore, they have been used extensively in gene expression profiling studies on the molecular mechanisms of metabolic disorders, including GDM [19]. 

## 2. Results

### 2.1. Clinical Data for the Groups Studied

This study cohort compromised a total 59 subjects—28 women with GDM at the time of GDM diagnosis (GDM group) and at the 1-year follow-up (pGDM group), and 31 pregnant women with NGT (NGT group). Among the pGDM subjects, 18 (64.3%) sustained NGT (pGDM-NGT) and 10 (35.7%) developed abnormal glucose tolerance (pGDM-AGT), which included 9 (32.1%) prediabetes women (8 IFG + 1 IGT) and 1 (3.6%) T2DM women. The procedure for assigning patients to the appropriate study groups during and after gestation is shown in Figure 1. Because the size of the pGDM-AGT and pGDM-NGT groups was too small, producing a result which was not sufficiently powered to detect a difference between the subgroups, all analyses in this study were made for the pGDM group as a whole. Detailed clinical characteristics of all the study groups are provided in Table 1. At GDM diagnosis, the GDM and NGT groups were closely matched for maternal age, gestational age, and the degree of whole adiposity (i.e., the indices of pre-pregnancy BMI, pregnancy weight, and gestational weight gain) (*p* > 0.05). The patients with GDM displayed significantly higher all-plasma glucose values following the OGTT, higher fasting insulin levels, higher HOMA-insulin resistance (HOMA-IR) indexes, and markedly lower quantitative insulin sensitivity check index (QUICKI-IS) and values in some lipid components (total cholesterol (TC), HDL-cholesterol (HDL-C), and LDL-cholesterol (LDL-C)) than the women with NGT. No significant differences for glycated hemoglobin (HbA1C), triglycerides (TGs), C reactive protein (CRP), and HOMA-B were observed between both groups (*p* > 0.05).

At the 1-year follow-up, the pGDM group exhibited significantly higher FPG, but lower 2 h post-load glucose, and lower measurements in all the lipid components (TC, LDL-C, HDL-C, and TGs), HOMA-β cell function (HOMA-B) index, and CRP, as compared with the NGT and GDM groups in pregnancy. Of note, the values of fasting insulin, HOMA-IR, and QUICKI-IS in the pGDM group were similar to those in the NGT group (*p* > 0.05). There was no significant difference in HbA1C values between the pGDM, GDM, and NGT groups, and postpartum BMIs returned to those before pregnancy (*p* > 0.05).

### 2.2. Gene Expression Alterations in Inflammation-Related Genes 

Using a qPCR-based approach, the expression changes of all the studied inflammation-related genes were quantified for the NGT, GDM, and pGDM groups, and these data are presented in Table 2 and Figure 2. The expression of *IL6* (FC 4.07, *p* = 0.0000, and FC 2.10, *p* = 0.0022), *IL18* (FC 3.21, *p* = 0.0049, and FC = 6.1, *p* = 0.0004), and *IL10* (FC 2.18, *p* = 0.0369, and FC 2.35, *p* = 0.0004) was upregulated both in the GDM and pGDM groups, respectively, as compared to the NGT group. When a comparison of the expression of these three cytokines was performed between the GDM and pGDM groups, only postpartum *IL18* transcript was markedly increased (FC = 1.89, *p* = 0.0382) in the pGDM group, while the *IL6* and *IL10* mRNA levels did not differ between the two groups (*p* > 0.05). The *IL8* expression profile showed a U-shape, with a significant decrease in the GDM group relative to the NGT group (FC 0.57, *p* = 0.0004) and a significant increase at postpartum follow-up compared to the GDM group (FC = 1.86, *p* = 0.0006), but achieved a level similar to that of the NGT group (*p* > 0.05). The postpartum mRNAs of *IL13* (FC = 5.30 and FC = 9.68, all *p* = 0.0000) and *RELA* (FC = 1.20, *p* = 0.0323, and FC = 1.58, *p* = 0.0443) were significantly upregulated, as compared to the GDM and NGT groups, respectively; however, their expression levels remained unchanged between the GDM and NGT groups in pregnancy (*p* > 0.05). No significant difference was observed in the *TNFA* expression profile between the GDM, pGDM, and NGT groups, although a tendency for a decreased *TNFA* level was seen in the pGDM group compared to the GDM group (FC = 0.69, *p* = 0.0795). 

### 2.3. Correlations between Gene Expression and Clinical Variables 

Correlations between the expression of the genes investigated and clinical phenotypes of the subjects during and after diabetic pregnancy were performed applying Spearman’s test with multiple testing corrections using the false discovery rate (FRD) method (Table 3 and Table 4). In GDM and NGT combined, *IL6* transcript was significantly and positively correlated with blood glucose measures, both at fasting (*r* = 0.42, *p* = 0.0010) and at 2 h OGTT (*r* = 0.39, *p* = 0.0022), whereas *RELA* transcript negatively correlated with HbA1C (*r* = −0.43, *p* = 0.0007). In the pGDM group, only *IL10* (*r* = 0.62, *p* = 0.0006) positively associated with plasma glucose concentration at 2 h during the OGTT. There were no other significant correlations between any of the other transcripts and clinical variables of the participants during and after pregnancy.

### 2.4. Gene–Gene Expression Correlations 

Individual relationships between the transcripts studied were defined during pregnancy and 1 year after childbirth, based on the Spearman correlation measures, after an FDR correction; the results are provided in Table 3 and Table 4. As illustrated in Figure 3, several shared gene–gene correlations were found between *IL18, IL6*, and *IL10* in the study periods; however, each period also had unique associations, such as *IL10* with *IL6,* and *RELA* and *IL18* with *TNFA*, for the entire group of pregnant women, and *IL8* with *IL10* and *IL18* for the pGDM group. Of note is that all gene–gene correlations were positive and moderate or high.

### 2.5. Diagnostic Potential of Inflammation-Related Genes

The receiver operating characteristic (ROC) curves were plotted, and the area under the ROC curves (AUCs) and optimal cut-off points were calculated to estimate diagnostic values for the expression of inflammation-related genes in the study population. As is evidenced in Table 5 and Figure 4, *IL6* (AUC 0.844, 95% CI 0.736–0.953, *p* = 0.0000, cut-off 33.314), *IL8* (AUC 0.771, 95% CI 0.651–0.890, *p* = 0.0000, cut-off 0.803), and *IL18* (AUC 0.714, 95% CI 0.582–0.846, *p* = 0.0014, cut-off 0.327) separately had satisfactory diagnostic value for GDM; therefore, it seems that the expression of each of these genes could correctly discriminate between GDM patients and those with NGT. In the case of *IL10*, the discriminative accuracy was rather poor (AUC 0.658, 95% CI 0.511–0.805, cut-off 0.485); thus, it may not be sufficient to use *IL10* as a single biomarker. ROC curve analyses for the combination of these four genes were also performed; however, no significant results were observed.

### 2.6. Prediction of Postpartum AGT Incidence

We further determined how well expression of the study genes can predict the development of AGT at 1-year postpartum in women with prior GDM. As shown in Table 6 and Figure 5, the AUC for *IL13*, *IL8*, and *TNFA* was 0.733 (95% CI 0.538–0.929, *p* = 0.0193), 0.750 (95% CI 0.53–0.970, *p* = 0.0260), and 0.761(95% CI 0.565–0.957, *p* = 0.0089), respectively. Thus, each of these genes had moderate accuracy for predicting the AGT state after diabetic pregnancy; however, when the expression of these genes was combined, the three-gene signature had a more significant predictive potential (AUC 0.85, 95% CI 0.689–1.000, *p* = 0.0000, cut-off 1.353) than each of these genes individually.

ROC curve analyses were also carried out for each maternal anthropometrical and biochemical parameter provided in Table 1. Of all the variables evaluated, only FPG could predict the development of AGT postpartum with a diagnostic accuracy (AUC 0.711, 95% CI 0.517–0.905, *p* = 0.0328) that was similar to that of *IL8*, *IL13*, or *TNFA*, and with an optimal cut-off of 101 mg/dL. The combination of FPG and the three-gene signature, compared to the three-gene signature alone, did not improve accuracy for the detection AGT postpartum (Table 6).

Subsequently, we constructed diverse logistic regression models based on the best predictors found in the ROC analyses (Appendix A). We analyzed the univariate models containing single raw variables, i.e., the expression of *IL8*, *IL13*, *TNFA*, or the concentration of FPG (the models 1–4), as well as the univariate models containing single predictors made up of linear combinations of variables (i.e., the models 9 (*IL8* × *IL13* × *TNFA*) and 10 (*IL8* × *IL13* × *TNFA* × FPG)) (Appendix A). We also analyzed the multivariable models which were additive models (i.e., the models 5 (*IL8* + *IL13* + *TNFA* + FPG), 6 (*IL8* + *IL13* + *TNFA)*, and 8 (*IL8* + *IL13*)), and the model with interaction (i.e., the model 7 (*IL8* × *TNFA* + *IL13)*) (Appendix A). The models were tested for overall significance of coefficients and fit to the data by specific statistics, and checked for overfit by v-fold cross validation. From the investigated models, the combination of the expression of *IL8*, *IL13,* and *TNFA* (LR test *p* = 0.0002, cross-validation AUC 0.800, 95% CI 0.5844–1.000) was the strongest predictor of development of AGT at 1-year postpartum among the GDM women (Table 7). Addition of FPG to this model did not improve its performance significantly (LR test *p* = 0.0001, cross-validation AUC 0.760, 95% CI 0.525–0.995). 

## 3. Discussion

This relatively long-term study included 31 healthy pregnant women and 28 patients with GDM, selected based on rigorous inclusion criteria. Of these, 10 (35.7%) developed AGT in the first year after delivery. Thus, the incidence of post-GDM glucose abnormality in the present study was higher than that reported by previous investigators (16.7% in [20] and over 25.5% [21]) during the first year post-delivery. It is now evident that the majority of women with GDM return to normal glycemic status after delivery; however, they are at increased risk for developing postpartum AGT, depending on the period of follow-up, the diagnostic criteria for pregnancy complicated by GDM, and characteristics of the studied population [22].

The present study included a homogenous population of the participants in regard to ethnicity, with no significant differences in maternal age, gestational week at the OGTT, or obesity parameters (i.e., pre-and post-pregnancy BMI, pregnancy weight, and gestational body weight gain), thereby eliminating effects of these factors on metabolic phenotypes and leukocyte expression profiles of inflammation-related genes of diabetic patients during and after pregnancy. 

Clinically, we found that the patients with GDM, compared with the healthy pregnant women, were hyperglycemic, with significantly higher fasting and post-load glucose concentrations, and hyperinsulinemic, reflecting more insulin-resistance and less insulin sensitivity in these patients, as evidenced by significantly higher HOMA-IR and lower QUICKI-IS indices, respectively. This result seems to parallel to previously reported findings [23,24]. At the 1-year follow-up, former GDM patients remained hyperglycemic, with significantly higher fasting but lower 2 h blood glucose levels, and displayed a significant decrease in HOMA-B, while HOMA-IR and QUICKI-IS were unchanged. Thus, it can be concluded that the key defect leading to the deterioration of hyperglycemia after diabetic pregnancy is impaired insulin secretion due to beta cell failure [25]. Of note is that all maternal lipids (i.e., TC, HDL-C, LDL-C, and TGs) decreased progressively from GDM to pGDM, achieving significantly lower levels at the postpartum period; however, they remained within the normal range [26]. Interestingly, the CRP level was similar in the GDM and NGT groups, and it decreased significantly from pregnancy to postpartum. The reason for this is unclear, but regarding the pregnancy period, it may be due to the fact that there were no differences in adiposity parameters between the GDM and NGT groups, as some researchers have previously described [27,28]. On the other hand, as pre-pregnancy BMI has emerged as the most important determinant of CRP concentration during pregnancy [28], it is tempting to hypothesize that the decreased CRP concentrations at follow-up in women with previous GDM may stem from the fact that they returned to their pre-pregnancy BMI < 25 kg/m^2^.

Despite existing evidence that low-grade systemic inflammation is an important feature in GDM patients during and after pregnancy [16,29], exact information about effects of the two states on leukocyte transcriptional responses in regard to several key pro- and anti-inflammatory genes is still lacking. We have filled this gap in our knowledge by showing a modified leukocyte transcriptional response in the GDM patients at the time of GDM diagnosis and 1 year after giving birth vs. normal pregnancy, with GDM-specific downregulation of *IL8* and pGDM-specific upregulation of *IL13* and *RELA*. These changes were accompanied by a significant upregulation of leukocyte *IL6*, *IL10*, and *IL18* mRNAs, both in GDM and pGDM groups, with the highest abundance being seen for postpartum *IL18* (FC = 6.1). Hence, these results suggest that both GDM and pGDM are closely linked to transcriptional dysregulation of certain pro-and anti-inflammatory genes in maternal leukocytes.

IL-8, also called C-X-C Motif Chemokine Ligand 8 (CXCL8), is a pro-inflammatory chemokine that participates in neutrophil activation and, thereby, is considered as a key player in the pathogenesis of some inflammatory diseases [30]. In the context of diabetes, a recently published meta-analysis indicated that the progression of T2DM can be linked to elevated concentrations of different chemokines, including IL-8 [31]. With respect to gestational diabetes, there are inconsistent results on alterations of this chemokine in GDM vs. NGT patients and, hence, its role in GDM remains unclear [32,33,34]. In the current study, the levels of leukocyte *IL8* mRNA were lower in the GDM patients than the NGT pregnancies; however, their postpartum levels returned to those observed in healthy pregnant subjects, indicating that the GDM patients are characterized by a unique leukocyte *IL8* expression pattern. Although a plausible explanation for this change is not available, a GDM-specific downregulation can probably be seen as protective in the context of inflammatory process, as this interleukin possesses a chemotactic activity for leukocytes and contributes to amplification of inflammation [30]. A similar conception has been proposed by Purohit et al. [35] in respect to IL-8 in type 1 diabetes mellitus (T1DM), based on the finding that circulating IL-8 levels were markedly decreased in patients with T1DM compared to normal control. Thus, further research is clearly required to understand the protective mechanism of action of IL-8, not only in patients with GDM but also in those with T1DM complications.

The pro-inflammatory cytokines, such as IL-18 and IL-6, and the anti-inflammatory cytokine IL-10, have been shown to be associated with GDM in several studies [36,37,38]; however, little is known at present about their changes in women with prior GDM. In the current study, leukocyte *IL6*, *IL10*, and *IL18* mRNAs were elevated in the GDM women during and after pregnancy, with significant positive correlations found for *IL18* with *IL6* and *IL10* in both study periods. It suggests the existence of highly co-regulated transcriptional overexpression of these genes in both study periods, which might be, at least in part, a result of the hyperglycemia that is evident in the GDM and pGDM groups. Furthermore, it appears that *IL6* and *IL10* transcripts contribute to glucose homeostasis in pregnancy and postpartum, respectively, as evidenced by positive correlations for these transcripts with glycemic variables in both periods using Spearman’s correlation analyses with FRD correction. In regard to IL-6, this finding is not surprising, since this cytokine, in addition to its function in the immune system, also participates in glucose metabolism during diabetic pregnancy [29], although it is not fully clear what the exact mechanism is by which IL-6 is linked to abnormal glucose homeostasis in GDM patients [39,40]. In the case of IL-10, the importance of its association with glycemia has been highlighted by a previous study in vitro, where enhanced IL-10 synthesis by human monocytes was observed in the presence of increasing concentrations of glucose [41]. Despite this, a mixed set of results has been reported on an association between circulating IL-10 levels and GDM, with findings showing unchanged, decreased, or increased IL-10 levels in diabetic pregnancies compared to healthy pregnant women [16,41]. Although these discrepancies are currently difficult to explain, a recent study has suggested that increased levels of circulating IL-10 in diabetic pregnant women might be a consequence of hypomethylation of the *IL10* gene caused by hyperglycemia [41]; however, whether a similar epigenetic mechanism could underlie an elevated *IL10* transcript in leukocytes of hyperglycemic GDM women during pregnancy and at 1-year follow-up remains to be determined. On the other hand, as IL-10 is thought to be a key suppressor of the immune response [42], the observed upregulation of this cytokine in both timepoints in our study might also be a compensatory protective mechanism in response to the overexpression of *IL6* and *IL18,* i.e., both cytokines with pro-inflammatory actions found in our hyperglycemic GDM women during and after pregnancy, and this possibility needs to be addressed by future studies. Of greater relevance, higher plasma IL-18 levels have been detected in response to acute hyperglycemia in healthy volunteers and subjects with impaired glucose tolerance, suggesting a direct link between IL-18 and elevated glucose levels [43]. 

Another observation captured in our study is that the *IL13* and *RELA* transcripts were remarkably upregulated in the pGDM group compared to the GDM and NGT groups, whereas we did not observe any significant difference between the GDM and NGT groups in relation to the aforementioned transcripts. It indicates that only women with prior GDM, characterized by hyperglycemia and impaired insulin secretion in our study, exhibited unique *IL13* and *RELA* overexpression profiles. IL-13 is an anti-inflammatory cytokine belonging to the alpha-helix protein family that has been implicated in the pathogenesis of parasitic infection, asthma, and allergic diseases [44], as well as in the metabolic regulation of glucose and insulin resistance [45,46], although clinical findings in humans concerning the role of IL-13 in metabolic diseases are still controversial [46,47]. To our knowledge, only one study has evaluated circulating IL-13 concentrations in women with GDM vs. normal pregnant women, where no significant difference in its level was found between the groups [48]. Hence, our data expand on this body of work by revealing that leukocyte *IL13* mRNA levels do not differ significantly between the women with GDM and those with NGT. Because human data investigating *IL13* expression in women with prior GDM are missing, it is difficult to explain why leukocyte *IL13* gene expression is significantly higher in the postpartum period compared with pregnancy in the present study. One could suggest that relatively high *IL13* levels observed in the women with prior GDM (FC = 5.30 and FC = 9.86 relative to the GDM and NGT groups, respectively) might counteract leukocyte *RELA* upregulation in these subjects (FC = 1.20 and FC = 1.58 relative to the GDM and NGT groups, respectively); however, critical appraisal of this hypothesis is required, since no correlation was evident between postpartum *IL13* and *RELA* transcripts. It is now well known that, as a member of NF-κB family, the p65 subunit encoded by the *RELA* gene is a critical regulator of cellular functions of NF-κB, such as pro-inflammatory gene induction and overproduction of reactive oxygen species (ROS), that play a pathogenic role in various inflammatory diseases [49,50]. Currently, very little evidence is available regarding the significance of NF-κB in women with prior GDM. In this sense, it has been previously shown that 3 months postpartum, women with a history of GDM had significantly higher expression of some genes involved in the activation of NF-κB [51]. Interestingly, although our statistical analysis failed to demonstrate a relationship between postpartum *RELA* expression and the clinical characteristics of patients, an inverse correlation of *RELA* transcript with maternal HbA1c level was found in the entire group of pregnant women at the time of GDM diagnosis, thus suggesting an enhanced *RELA* response in pregnancies with a decrease of HbA1c levels. Contrary to our finding, Hofmann et al. [52] noted increased NF-κB activation in circulating mononuclear cells in patients with poorly controlled diabetes. Likewise, a further study demonstrated that glucose normalization results in a reduction of NF-κB activation in mononuclear blood cells of patients with T1DM [53]. Taking into account these conflicting results, future studies are required to specifically explore how the late phase of maternal hyperglycemia in pregnancy might affect leukocyte *RELA* expression.

Another finding that emerges from the analysis of this study is unchanged *TNFA* expression from pregnancy to a year post-birth, although a trend for lower postpartum *TNFA* compared to the GDM group was noted. Complementary to this, some studies have also shown no alterations of circulating TNF-α levels in pregnant women with GDM compared to non-diabetic subjects [37,54]; however, other studies have not confirmed such findings, revealing elevated levels of this cytokine in pregnancies with GDM [55]. Given that TNF-α is currently considered as one of the most important factors of insulin resistance in diabetes mellitus, including GDM [56,57,58], the observed unaltered levels of *TNFA* in the GDM women vs. pregnancies with NGT was somewhat surprising, despite higher HOMA-IR and lower QUICKI-IS indices found in the diabetic subjects. The reason for this inconsistency remains unknown. The lack of changes in major parameters of obesity, such as pre-pregnancy BMI and pregnancy weight, between the GDM and NGT groups might contribute to some degree, according to the results reported by some, but not all, previous studies [59]. In line with this concept, we noticed a positive correlation between *TNFA* transcript and the above-mentioned parameters of obesity in the entire group of pregnant women, which disappeared when an FDR correction was applied. Another possible explanation is that leukocytes might be a less important source of circulating TNF-α than adipose tissue or the placenta in the context of the pathogenesis of insulin resistance in diabetic pregnancy [60]; however, further research is needed to verify the aforementioned hypotheses and explore how leukocyte-derived TNF-α functions in women with prior GDM.

We also attempted to identify potential diagnostic biomarkers among the investigated transcripts that differentiate pregnant women with and without GDM. Although OGTT is currently the gold standard for GDM diagnosis [61], it is time-consuming and uncomfortable for some pregnant women, inducing nausea and vomiting [62]. Additionally, the diagnosis of GDM based on blood glucose values is still controversial, with different international diagnostic criteria used. Hence, the identification of new biomarkers for diagnosing GDM, without performing OGTT, is a pending task. In our study, considering each transcript separately, promising diagnostic potential to distinguish GDM patients from healthy controls in the time of GDM diagnosis was found for *IL6* (AUC 0.844), *IL8* (AUC 0.771) and *IL18* (AUC 0.714), whereas *IL10* does not appear to be a fully satisfactory biomarker on its own (AUC 0.658). Thus, our data suggest that leukocyte gene expression of *IL6*, *IL8*, or *IL18* can provide an alternative non-invasive approach to distinguish between pregnant women with and without GDM who are closely matched for maternal age, gestational age, and parameters of obesity such as pre-pregnancy BMI, pregnancy weight, and gestational weight.

As the early prediction of postpartum AGT among individuals with GDM is of great importance for the application of appropriate preventative actions, we also searched for predictive transcriptomic and clinical biomarkers for postpartum AGT among our diabetic patients. For this purpose, numerous logistic regression models and ROC curves were constructed and compared. From the investigated models, the combination of *IL8*, *IL13*, and *TNFA* transcripts provided the best prediction model for detecting the cases developing AGT at 1-year postpartum among the GDM women. Of note is that the combination of the three-gene signature with FPG did not improve its predictive value, confirming that a three-gene panel was driving the prediction model.

The strength of this study is the follow-up design with detailed anthropometrical and metabolic measures of the patients at the time of GDM diagnosis and 1-year postpartum. Furthermore, the pregnancies with and without GDM were closely matched for ethnicity, maternal age, gestational week at the OGTT, as well as for obesity parameters, including pre-and post-pregnancy BMI, pregnancy weight, and gestational body weight gain. Despite this, the small sample size of the study and the lack of a group of postpartum women without a history of GDM are the main limitations of our study. This is due to the poor postpartum response to recall for follow-up, a problem observed by others [63,64]. Moreover, we did not possess information on the diet of the subjects and their physical activity, i.e., factors that may affect the expression of numerous inflammation-related genes [65,66]. Finally, only Caucasian pregnant women were included in this study and our findings may, therefore, not apply to other ethnic groups.

In conclusion, our data demonstrate a differential transcriptional response of the set of inflammation-related genes in leukocytes of the pregnancies with GDM at the time of GDM diagnosis and 1-year postpartum, with GDM-specific downregulation of *IL8* and pGDM-specific upregulation of *IL13* and *RELA* accompanied by upregulation of *IL6*, *IL10*, and *IL18* in both groups. Our results also indicate that the expression of *IL6*, *IL8*, or *IL18* alone has a satisfactory capability to distinguish the GDM patients from healthy pregnancies at the time of GDM diagnosis; thus, each of these genes possesses the potential to serve as a good molecular biomarker helping to identify GDM patients. Furthermore, our study provides the three-gene signature, consisting of *IL8*, *IL13*, and *TNFA*, for identifying GDM individuals at high risk of developing AGT postpartum, which might help design appropriate interventions and implement health policies that may aid in AGT prevention. However, further studies in larger cohorts with different ethnic groups are necessary to validate the results of the present study. These studies should also be designed to include postpartum women following a normal pregnancy.

Since literature exists showing that hyperglycemia, the advanced glycation end-products (AGEs)/advanced glycation end-product receptor (RAGE) axis, oxidative stress, and immune activation are integral interconnected pathological features in diabetes including GDM [67,68], further research in this field should be carried out in the patients enrolled here. Such a comprehensive approach has the potential to further our understanding of the progression from GDM to postpartum AGT and provide a simple and accurate alternative method to assess the risk of future AGT in women with GDM. 

## 4. Materials and Methods

### 4.1. Study Subjects

A total of 59 Caucasian pregnant women, including 28 women with GDM and 31 women with NGT, from the Outpatient Department of Diabetology, Lodz, Poland, were enrolled. All the pregnant women were routinely screened with a 75 g OGTT, and GDM was diagnosed when one or more whole-blood glucose values met or exceeded 92 mg/dL (5.1 mmol/L) for fasting, 180 mg/dL (10.0 mmol/L) for 1 h post-glucose load, and 153 mg/dL (8.5 mmol/L) for 2 h post-glucose load, according to the criteria set by the Polish Diabetes Association (PDA), based on the WHO/IADPSG guidelines [69,70]. The pregnant women with NGT had a negative screen and were relatively matched with GDM women for maternal age, gestational age at the OGTT, and parameters of obesity (i.e., pre-pregnancy BMI, pregnancy weight, and gestational weight gain), as evidenced by non-significant differences in these variables between the GDM and NGT groups (Table 1). The diabetic pregnant women were analyzed at the GDM diagnosis and, therefore, they had not received any therapy at the time of inclusion into the study. 

Women aged <18 or >35 years, with multiple pregnancies, autoimmune disease, any form of pre-pregnancy diabetes, and a respiratory infection or inflammatory illness were ineligible for the study.

Women diagnosed with GDM (*n* = 28) had a further OGTT at 1-year postnatal. Postpartum prediabetes (IFG or/and IGT) and T2DM were defined based on the 2014 PDA criteria [71], either through fasting glucose or after the 2 h value of the OGTT. The values between 100–125 mg/dL in fasting and 140 mg/dL–199 mg/dL in the OGTT confirmed IFG and IGT, respectively. T2DM was recognized when the value of blood glucose was equal to or greater than 200 mg/dL at 2 h of the OGTT. 

### 4.2. Clinical and Biochemical Measurements 

All patients underwent clinical and laboratory assessment at the time of diagnosing GDM and 1-year postpartum. Gestational age was calculated according to the date of the last menstrual period and verified by ultrasonography. Maternal height and weight were measured at the time of GDM diagnosis and patients provided information on their pre-pregnancy weight. The pre-pregnancy BMI (BMI = (weight (kilograms)/height m^2^)) and weight gain were calculated. Biochemical parameters of the patients, including plasma concentrations of FPG, HbA1c, TC, LDL-C, and HDL-C, TGs, fasting insulin, and CRP were determined after overnight fasting by routine laboratory examinations, according to the procedures described previously [72,73]. The HOMA-IR and HOMA-B indices were calculated as follows [74]:HOMA-IR = [fasting insulin (µU/mL) × fasting glucose (mg/dL)]/405;
HOMA-B = [360 × fasting insulin (µU/mL)]/[fasting glucose (mg/dL) − 63].

The QUICKI-IS was calculated using the following formula [75]: QUICKI-IS = 1/[log(I0) + log(G0)]
where I0 is the fasting plasma insulin level (µU/mL) and G0 is the fasting blood glucose level (mg/dL).

### 4.3. Leukocytes’ Separation and Total RNA Extraction

Fasting blood samples (10 mL) were collected from all participants during and after pregnancy to determine the aforementioned biochemical parameters and isolate leukocytes by adding red blood cell lysis buffer (0.5 M NH_4_Cl, 10 mM KHCO_3_, 1 mM EDTA, pH 8.0) followed by centrifuging at 4000 rpm for 10 min at 4 °C. Leukocytes were obtained after supernatant was discarded. Total RNA from maternal blood leukocytes was then extracted using commercially available acid-phenol reagent according to manufacturer’s protocol (Tri Reagent, Sigma-Aldrich, St. Louis, MO, USA). The concentration and purity of RNA were determined by the spectrophotometric method at wavelength λ = 260 nm and λ = 280 nm using a BioDrop UV/VIS Spectrophotometer (SERVA, Heidelberg, Germany).

### 4.4. cDNA Synthesis and Nested Quantitative Polymerase Chain Reaction (qPCR)

In order to obtain cDNA, total RNA (4 µg) was reverse transcribed with the use of the Thermo Scientific Maxima Reverse Transcriptase kit, according to the manufacturer’s recommendation (Thermo Scientific, Waltham, MA, USA). To enhance the sensitivity of the detection of the genes investigated in leukocytes of individuals, nested PCR was applied. This method consists of two steps of amplification: the first includes amplifying the target sequence with the outer primers, whereas the second utilizes the first PCR product as template for the inner primers for amplification [76]. For each gene analyzed, specific pairs of outer and inner primers were designed using the online NCBI Primer-Blast program (https://www.ncbi.nlm.nih.gov/tools/primer-blast, accessed on 15 August 2020) (Appendix A) [77]. The first round of amplification was performed in 20 μL reaction mix containing 10 μL Premix DFS-*Taq* DNA Polymerase (Bioron, Römerberg, Germany), 0.5 μM for each primer, and 2 μL of template cDNA (diluted 20×). The temperature profile for amplification was as follows: initial denaturation at 95 °C for 3 min, denaturation at 95 °C for 30 s, annealing at 58–62 °C for 30 s (depending on the tested gene), and extension at 72 °C for 20 s, for 15–25 cycles (depending on the tested gene), followed by a final extension at 72 °C for 3 min. To optimize the annealing temperature for primer pairs, gradient temperature PCR for each amplicon was performed. The second round of amplification was carried out using 2 μL of the first PCR product (diluted 200×) as the template, and using the inner primers of the sequences provided in Appendix A and GoTaq^®^ qPCR Master Mix (2×) reaction kit (Promega, Madison, WI, USA), according to manufacturer’s recommendations. Reactions were performed in duplicate on a LightCycler 480 II (Roche Diagnostics GmbH, Basel, Switzerland) with initial denaturation at 95 °C for 2 min, followed by 40 cycles of 95 °C for 20 s and 60 °C for 20 s. The amplification of specific transcript was confirmed by melting curve at the end of each PCR and electrophoresis on 1.5% agarose gel in Tris-Borate-EDTA (TBE) buffer. The *ACTB* gene encoding β-actin was used as the housekeeping gene for internal normalization. The relative expression of the target genes was calculated using the threshold cycle (C_t_) value according to the Pfaffl method [78]. Of note, RNA without reverse transcriptase during cDNA synthesis as well as PCR reaction using water instead of template showed no amplification. 

### 4.5. Statistical Analyses

Data are presented as median with interquartile range (IQR). The distribution of clinical and expression data was measured by the Shapiro–Wilk test. Differences between the groups studied were compared by the nonparametric Mann–Whitney *U* test. The Wilcoxon matched-pairs signed-rank test was used to assess differences in the matched pairs data of the patients at different time intervals during the research. The non-parametric Spearman’s rank test was used for analysis of correlation between variables. Correlations were corrected for multiple testing by the Benjamini–Hochberg false discovery rate (FDR) method [79]. The ROC curves and the AUCs were determined to discriminate between patients with and without glycemic abnormalities at the time of GDM diagnosing and 1-year postpartum, and to assess diagnostic and prognostic accuracies of the individual and combined effects of the investigated genes, respectively. The cut-off values with the largest Youden’s index (sensitivity + specificity − 1) were used to determine the best cut-off expression values of the studied genes. The AUCs of ROC curves were tested by DeLong’s method. For each statistical analysis, 95% confidence interval (95% CI) was computed based on the bootstrap bias-corrected and accelerated method (bca) with 1000 replicates. The dependence of the risk of AGT at 1-year postpartum on predictors measured at pregnancy was assessed by univariate and multivariable logistic regressions. The obtained models were rated according to their significance and fit to the data. The significance of coefficients of the models was assessed by likelihood ratio test (LR test) and Wald’s test. The fit of the models to the data was estimated by Hosmer–Lemeshow’s test, Akaike Information Criterion (AIC), Akaike Information Corrected Criterion (AICC), Bayesian Information Criterion (BIC), and the AUC of ROC curve. Each model was generated with accompanying v-fold cross validation to prevent overfit. The AUC for ROC curves of models for cross-validation data were calculated. On the basis of all the calculated statistics, the best model was indicated and estimates with standard error, as well as odds ratio with 95% confidence interval, were calculated. The significance of the estimates of the model was tested with Wald’s test. Statistical analyses were carried out using a commercially available statistical software package (Statistica version 12.5, StatSoft, Poland), and statistical significance was set at *p* < 0.05.

## Figures and Tables

**Figure 1 ijms-23-14677-f001:**
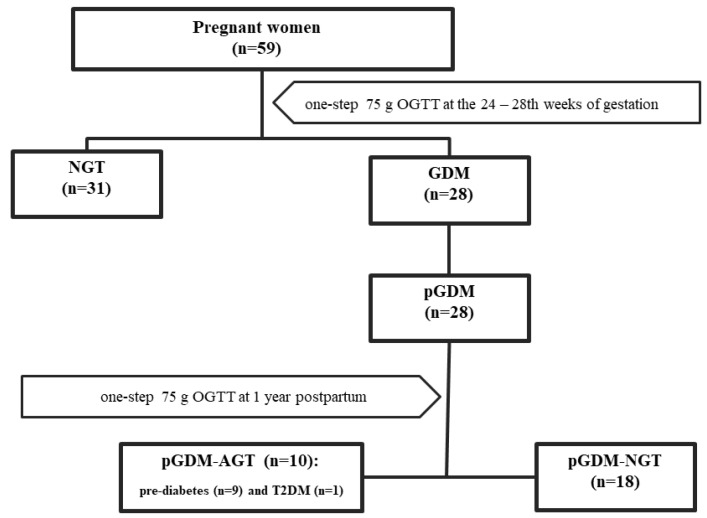
The classification of the study subjects during and after pregnancy based on the OGTT results.

**Figure 2 ijms-23-14677-f002:**
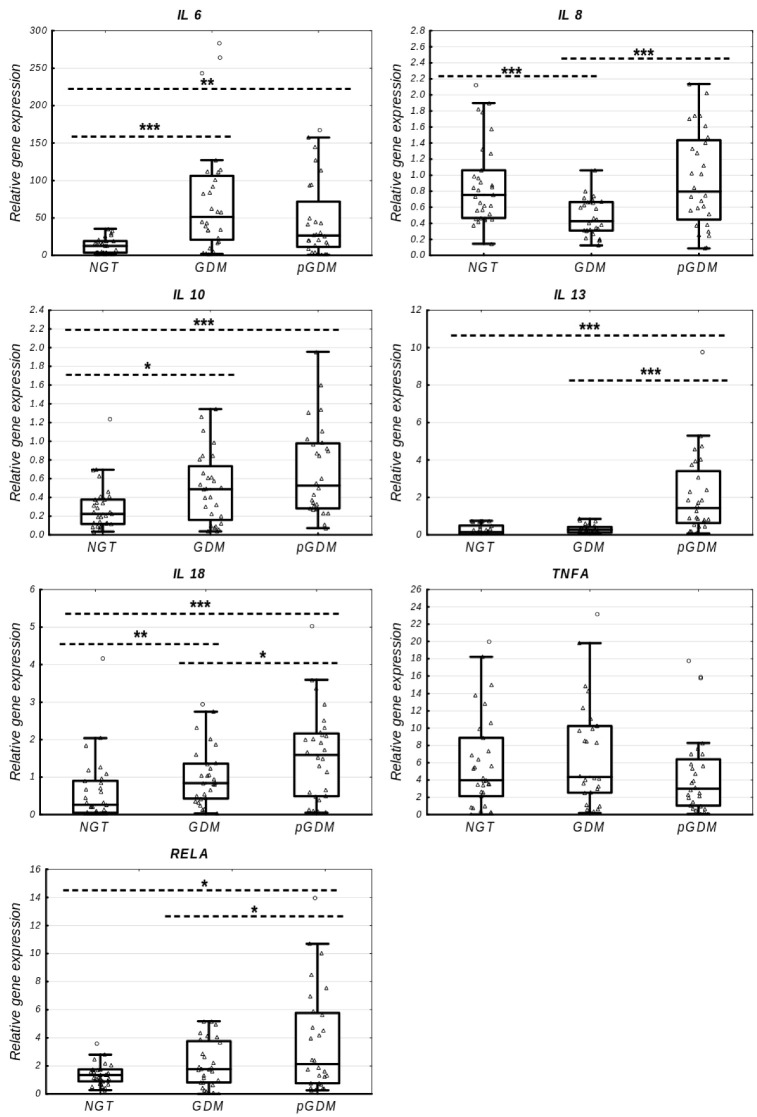
Comparison of the relative leukocyte mRNA expression (the ratio of the target gene relative to the reference gene ACTB) of all genes investigated between the NGT (*n* = 31), GDM (*n* = 28), and pGDM (*n* = 28) groups. Data are expressed as median (indicated by horizontal bars) with interquartile range (indicated by boxes) and 1.5 interquartile ranges (indicated by Tukey whiskers). Raw data points are shown as triangles. Outliers are marked as circles. * *p* < 0.05, ** *p* < 0.01, *** *p* < 0.001 as assessed by the Mann–Whitney *U* test.

**Figure 3 ijms-23-14677-f003:**
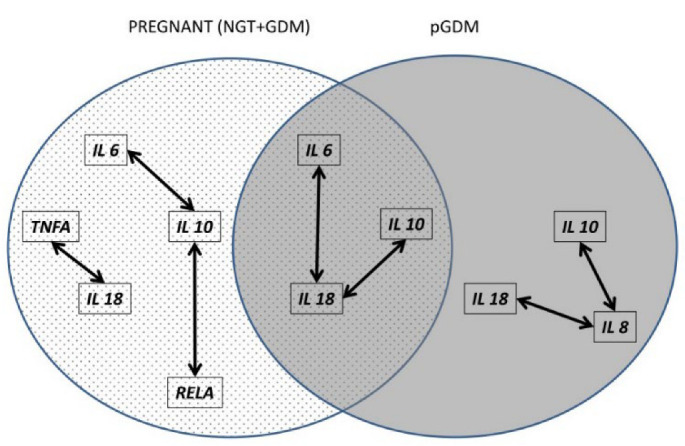
Scheme illustrating distinct and overlapping sets of gene–gene positive correlations throughout pregnancy and 1 year after delivery.

**Figure 4 ijms-23-14677-f004:**
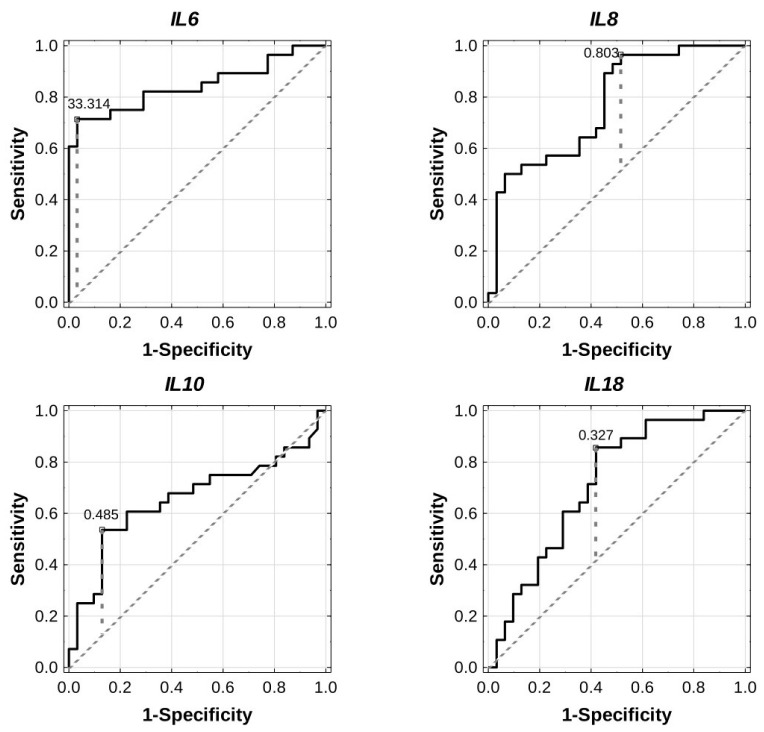
ROC curve analyses for differential expressions of the four investigated genes (*IL6*, *IL8*, *IL10*, and *IL18*) to discriminate GDM patients from normal controls. Solid lines represent the performance of each classifier. Diagonal dotted lines represent a reference classifier with the random performance level (AUC = 0.5). The optimal cut-off point (solid circle in white) for the expression of each gene is designated by the Younden index method (vertical dashed lines). The calculated results based on ROC analyses are available in Table 5.

**Figure 5 ijms-23-14677-f005:**
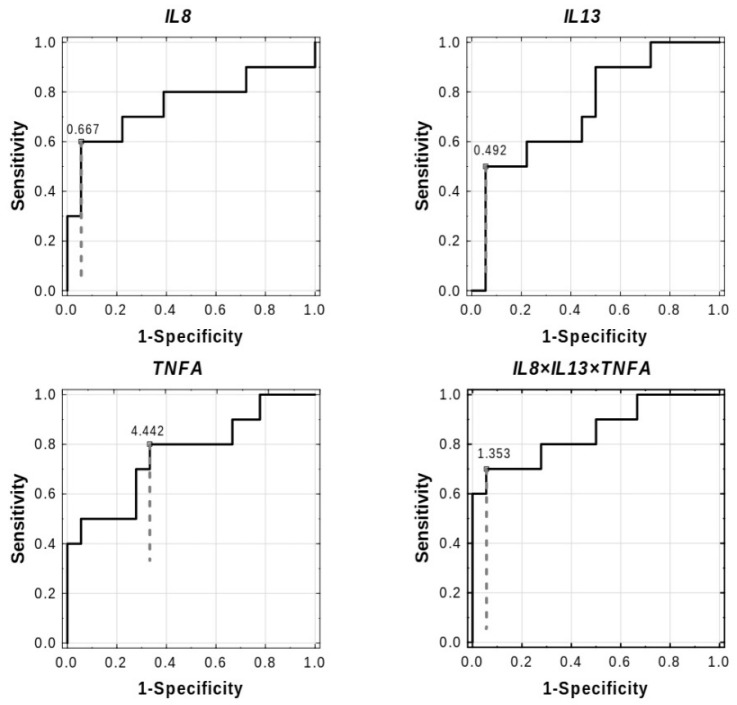
ROC curve analyses for the expression of the three genes (*IL8*, *IL13*, and *TNFA*), individually and in combination, for predicting AGT at 1-year postpartum in women with prior GDM. Solid lines represent the performance of each classifier. Diagonal dotted lines represent a reference classifier with the random performance level (AUC = 0.5). The optimal cut-off point (solid circle in white) for the expression of each gene is designated by the Younden index method (vertical dashed lines). The calculated results based on ROC analyses are available in Table 6.

**Table 1 ijms-23-14677-t001:** Baseline characteristics of the study groups.

	Pregnancy			Postpartum		
Variable	NGT (*n* = 31)	GDM (*n* = 28)	GDM vs. NGT *p*	pGDM (*n* = 28)	pGDM vs. GDM *p*	pGDM vs. NGT *p*
Maternal age (years)	30.00 (26.00–31.00)	30.50 (28.00–34.00)	0.2363	-	-	-
Gestational age at OGTT (week)	25.00 (24.00–26.00)	25.00 (24.00–26.00)	0.6309	-	-	-
Pre-pregnancy BMI (kg/m^2^)	23.29 (20.84–27.15)	25.30 (22.92–28.23)	0.1871	-	-	-
Pregnancy weight (kg)	71.35 (62.70–81.70)	76.40 (67.80–85.00)	0.1817	-	-	-
Gestational weight gain (kg)	8.00 (5.40–9.85)	9.15 (6.05–11.05)	0.3717	-	-	-
Postpartum BMI (kg/m^2^)	-	-	-	24.74 (21.85–30.51)	0.1068 #	0.2879 #
FPG (mg/dL)	81.00 (74.00–85.00)	86.50 (81.00–101.00)	0.0024 *	93.50 (86.00–101.50)	0.0400 *	0.0000 *
1-h OGTT (mg/dL)	159.00 (144.00–170.00)	182.00 (161.00–189.00)	0.0012 *	-	-	-
2-h OGTT (mg/dL)	137.00 (125.00–147.00)	158.00 (147.00–170.50)	0.0000 *	102.00 (89.00–108.00)	0.0000 *	0.0000 *
HbA1C (%)	5.29 (5.02–5.49)	5.09 (4.88–5.34)	0.2709	5.11 (4.96–5.29)	0.9595	0.1311
Fasting insulin (μIU/mL)	8.05 (6.50–9.80)	12.60 (7.60–16.30)	0.0223 *	9.05 (5.30–11.50)	0.0012 *	0.5038
HOMA-B	156.25 (122.40–183.00)	187.26 (138.09–242.96)	0.3445	104.21 (81.18–168.00)	0.0019 *	0.0398 *
HOMA-IR	1.63 (1.34–1.85)	2.66 (1.61–3.74)	0.0065 *	1.84 (1.13–3.07)	0.0119 *	0.3308
QUICKI-IS	0.15 (0.15–0.16)	0.14 (0.14–0.15)	0.0065 *	0.15 (0.14–0.16)	0.0049 *	0.3308
TC (mg/dL)	271.35 (241.00–296.72)	242.55 (209.20–268.15)	0.0124 *	197.50 (167.00–209.00)	0.0000 *	0.0000 *
HDL-C (mg/dL)	84.35 (71.78–98.00)	73.05 (60.75–86.70)	0.0465 *	58.00 (48.00–75.00)	0.0001 *	0.0001 *
LDL-C (mg/dL)	144.50 (126.00–170.00)	123.50 (99.00–153.50)	0.0304 *	105.00 (88.00–133.50)	0.0123 *	0.0005 *
TGs (mg/dL)	212.95 (182.60–240.00)	210.60 (169.10–246.10)	0.5503	91.50 (50.85–124.95)	0.0001 *	0.0000 *
CRP (mg/dL)	3.16 (1.80–7.33)	3.72 (1.93–5.47)	0.8307	1.41 (1.00–2.34)	0.0062 *	0.0071 *

Abbreviations: BMI, body mass index; CRP, C reactive protein; FPG, fasting plasma glucose; HDL, high-density lipoprotein; HOMA-B, homeostasis model assessment of β-cell function; HOMA-IR, homeostasis model assessment of insulin resistance; LDL, low density lipoprotein; QUICKI-IS, quantitative insulin sensitivity check index; TC, total cholesterol; TGs, triglycerides. Data are presented as median and interquartile range (IQR); * *p* < 0.05 NGT vs. GDM and NGT vs. pGDM as assessed by the Mann–Whitney *U* test; GDM vs. pGDM and # postpartum BMI vs. pre-pregnancy BMI as assessed by the Wilcoxon matched-pairs signed-rank test for paired results.

**Table 2 ijms-23-14677-t002:** Relative mRNA expression and fold changes (FC) of individual genes in leukocytes of women during and after pregnancy.

	Pregnancy		GDM vs. NGT		1-Year Postpartum	pGDM vs. GDM		pGDM vs. NGT	
Variable	NGT (*n* = 31)	GDM (*n* = 28)	FC	*p*	pGDM (*n* = 28)	FC	*p*	FC	*p*
*IL6*	12.63 (3.59–19.22)	51.40 (20.85–106.19)	4.07	0.0000 *	26.50 (11.30–71.80)	0.52	0.1514	2.10	0.0022 *
*IL8*	0.75 (0.47–1.06)	0.43 (0.31–0.66)	0.57	0.0004 *	0.80 (0.45–1.44)	1.86	0.0006 *	1.06	0.7905
*IL10*	0.22 (0.12–0.38)	0.49 (0.16–0.73)	2.18	0.0369 *	0.53 (0.28–0.98)	1.08	0.1329	2.35	0.0004 *
*IL13*	0.15 (0.02–0.49)	0.27 (0.12–0.42)	1.85	0.1649	1.43 (0.63–3.41)	5.30	0.0000 *	9.68	0.0000 *
*IL18*	0.26 (0.05–0.90)	0.84 (0.43–1.36)	3.21	0.0049 *	1.59 (0.49–2.17)	1.89	0.0382 *	6.1	0.0004 *
*TNFA*	3.97 (2.13–8.88)	4.34 (2.52–10.24)	1.09	0.4990	2.99 (1.03–6.39)	0.69	0.0795	0.75	0.3055
*RELA*	1.35 (0.90–1.75)	1.77 (0.82–3.76)	1.31	0.1234	2.13 (0.77–5.77)	1.20	0.0323 *	1.58	0.0443 *

Data are presented as median and interquartile range (IQR); * *p* < 0.05 NGT vs. GDM and NGT vs. pGDM as assessed by the Mann–Whitney *U* test; GDM vs. pGDM as assessed by the Wilcoxon matched-pairs signed-rank test for paired results.

**Table 3 ijms-23-14677-t003:** Correlation study through Spearman’s linear regression analysis at GDM diagnosis (*n* = 59).

	*IL6*		*IL8*		*IL10*		*IL13*		*IL18*		*TNFA*		*RELA*	
Variable	*r*	*p*	*r*	*p*	*r*	*p*	*r*	*p*	*r*	*p*	*r*	*p*	*r*	*p*
Maternal age (years)	0.18	0.1975	−0.03	0.8335	0.16	0.2335	−0.25	0.0604	0.24	0.0728	−0.03	0.8003	−0.02	0.9067
Pre-pregnancy BMI (kg/m^2^)	0.04	0.7870	−0.15	0.2777	−0.01	0.9534	−0.05	0.7123	0.01	0.9410	0.30	0.0244	0.05	0.7394
Pregnancy weight (kg)	0.05	0.7015	−0.08	0.5611	−0.02	0.9065	0.01	0.9520	0.00	0.9809	0.30	0.0229	0.11	0.4242
Gestational weight gain (kg)	0.01	0.9358	0.08	0.5570	−0.10	0.4504	−0.12	0.3752	−0.10	0.4727	−0.08	0.5619	0.00	0.9774
FPG (mg/dL)	0.42	**0.0010 ***	−0.15	0.2722	0.22	0.0990	0.13	0.3264	0.25	0.0605	0.26	0.0454	0.22	0.0944
1-h OGTT (mg/dL)	0.32	0.0207	−0.06	0.6473	0.04	0.7639	−0.01	0.9660	0.16	0.2614	0.14	0.3231	−0.09	0.5322
2-h OGTT (mg/dL)	0.39	**0.0022 ***	−0.21	0.1064	0.27	0.0374	−0.10	0.4575	0.29	0.0275	0.13	0.3120	0.08	0.5353
HbA1C (%)	−0.19	0.1439	0.18	0.1737	−0.19	0.1587	−0.29	0.0297	−0.15	0.2547	−0.04	0.7454	−0.43	**0.0007 ***
Fasting insulin (μIU/mL)	0.13	0.4047	−0.13	0.3898	−0.13	0.4073	0.04	0.7860	−0.17	0.2624	0.15	0.3279	0.04	0.8140
HOMA-B	0.01	0.9261	−0.10	0.4971	−0.21	0.1716	−0.06	0.6785	−0.05	0.7205	0.08	0.6143	−0.04	0.7732
HOMA-IR	0.20	0.1803	−0.16	0.2991	−0.05	0.7339	0.06	0.6755	−0.13	0.3921	0.15	0.3236	0.08	0.6037
QUICKI-IS	−0.20	0.1803	0.16	0.2991	0.05	0.7339	−0.06	0.6755	0.13	0.3921	−0.15	0.3236	−0.08	0.6037
TC (mg/dL)	−0.23	0.1018	0.28	0.0418	−0.18	0.1844	−0.10	0.4818	−0.17	0.2115	−0.13	0.3402	−0.20	0.1562
HDL-C (mg/dL)	0.03	0.8357	0.04	0.7905	−0.13	0.3514	−0.15	0.2642	−0.01	0.9575	−0.20	0.1418	−0.21	0.1278
LDL-C (mg/dL)	−0.22	0.1067	0.31	0.0209	−0.10	0.4924	−0.02	0.9033	−0.12	0.3834	−0.07	0.6334	−0.10	0.4530
TGs (mg/dL)	−0.26	0.0550	0.12	0.3909	−0.21	0.1361	−0.13	0.3439	−0.36	0.0079	0.05	0.7382	−0.03	0.8402
CRP (mg/dL)	−0.18	0.2408	0.00	0.9847	0.08	0.5882	0.05	0.7537	0.20	0.1786	0.16	0.2769	−0.05	0.7593
*IL6*	-	-	−0.18	0.1745	0.48	**0.0001 ***	0.10	0.4424	0.72	**0.0000 ***	0.33	0.0110	0.26	0.0490
*IL8*	−0.18	0.1745	-	-	0.24	0.0685	−0.19	0.1406	−0.13	0.3342	−0.04	0.7587	0.20	0.1272
*IL10*	0.48	**0.0001 ***	0.24	0.0685	-	-	0.14	0.2857	0.61	**0.0000 ***	0.23	0.0789	0.58	**0.0000 ***
*IL13*	0.10	0.4424	−0.19	0.1406	0.14	0.2857	-	-	0.22	0.0924	0.24	0.0729	0.24	0.0636
*IL18*	0.72	**0.0000 ***	−0.13	0.3342	0.61	**0.0000 ***	0.22	0.0924	-	-	0.45	**0.0004 ***	0.28	0.0288
*TNFA*	0.33	0.0110	−0.04	0.7587	0.23	0.0789	0.24	0.0729	0.45	**0.0004 ***	-	-	0.10	0.4516
*RELA*	0.26	0.0490	0.20	0.1272	0.58	**0.0000 ***	0.24	0.0636	0.28	0.0288	0.10	0.4516	-	-

* Significant correlations (*p* < 0.05) after false discovery rate (FDR) correction are marked in bold. Abbreviations are indicated in Table 1.

**Table 4 ijms-23-14677-t004:** Correlation study through Spearman’s linear regression analysis at 1-year postpartum (*n* = 28).

	*IL6*		*IL8*		*IL10*		*IL13*		*IL18*		*TNFA*		*RELA*	
Variable	*r*	*p*	*r*	*p*	*r*	*p*	*r*	*p*	*r*	*p*	*r*	*p*	*r*	*p*
Postpartum BMI (kg/m^2^)	0.26	0.1920	0.32	0.1114	0.42	0.0332	0.31	0.1249	0.21	0.3010	0.25	0.2182	0.32	0.1139
FPG (mg/dL)	0.18	0.3578	0.12	0.5383	0.27	0.1702	−0.19	0.3232	0.21	0.2754	0.04	0.8214	0.36	0.0567
2-h OGTT (mg/dL)	0.07	0.7244	0.42	0.0312	0.62	**0.0006 ***	0.21	0.2908	0.31	0.1097	0.10	0.6278	0.22	0.2638
HbA1C (%)	−0.06	0.7737	0.22	0.2677	0.09	0.6560	−0.07	0.7231	−0.12	0.5458	0.02	0.9097	−0.08	0.6826
Fasting insulin (μIU/mL)	0.00	0.9894	−0.05	0.8215	0.12	0.5558	0.17	0.4161	−0.07	0.7249	0.19	0.3460	0.14	0.5062
HOMA-B	−0.06	0.7869	−0.10	0.6435	−0.14	0.5115	0.34	0.0963	−0.19	0.3679	0.26	0.2171	−0.08	0.6956
HOMA-IR	0.06	0.7729	−0.03	0.8868	0.19	0.3551	0.11	0.6110	−0.01	0.9593	0.21	0.3210	0.19	0.3730
QUICKI-IS	−0.06	0.7729	0.03	0.8868	−0.19	0.3551	−0.11	0.6110	0.01	0.9593	−0.21	0.3210	−0.19	0.3730
TC (mg/dL)	0.04	0.8389	0.33	0.0835	0.31	0.1086	−0.15	0.4597	0.18	0.3488	0.11	0.5754	0.00	0.9945
HDL-C (mg/dL)	−0.21	0.2861	−0.03	0.8758	−0.29	0.1380	0.03	0.8758	−0.10	0.6198	−0.33	0.0881	−0.09	0.6365
LDL-C (mg/dL)	0.04	0.8432	0.27	0.1638	0.38	0.0436	0.04	0.8259	0.17	0.3940	0.16	0.4027	0.02	0.9284
TGs (mg/dL)	0.03	0.8879	−0.01	0.9427	0.28	0.1562	−0.21	0.2729	−0.04	0.8487	−0.09	0.6476	0.10	0.6179
CRP (mg/dL)	0.41	0.0361	0.24	0.2304	0.17	0.3871	0.25	0.2060	0.34	0.0821	0.44	0.0208	0.05	0.8161
*IL6*	-	-	0.33	0.0814	0.41	0.0320	0.09	0.6437	0.69	**0.0001 ***	0.44	0.0194	0.27	0.1676
*IL8*	0.33	0.0814	-	-	0.59	**0.0009 ***	0.15	0.4547	0.61	**0.0005 ***	0.23	0.2358	0.44	0.0183
*IL10*	0.41	0.0320	0.59	**0.0009 ***	-	-	0.15	0.4581	0.66	**0.0001 ***	0.30	0.1154	0.35	0.0640
*IL13*	0.09	0.6437	0.15	0.4547	0.15	0.4581	-	-	0.09	0.6557	0.26	0.1815	0.06	0.7609
*IL18*	0.69	**0.0001 ***	0.61	**0.0005 ***	0.66	**0.0001 ***	0.09	0.6557	-	-	0.41	0.0288	0.47	0.0115
*TNFA*	0.44	0.0194	0.23	0.2358	0.30	0.1154	0.26	0.1815	0.41	0.0288	-	-	0.03	0.8770
*RELA*	0.27	0.1676	0.44	0.0183	0.35	0.0640	0.06	0.7609	0.47	0.0115	0.03	0.8770	-	-

* Significant correlations (*p*  <  0.05) after false discovery rate (FDR) correction are marked in bold. Abbreviations are indicated in Table 1.

**Table 5 ijms-23-14677-t005:** The validity of gene expression in discriminating pregnant GDM and NGT women at GDM diagnosis (*n* = 59).

Gene	AUC	SE	95% CI	*p*	Effect	Cut-Off Value	95% CI	YI	95% CI
*IL6*	0.844	0.055	0.736–0.953	**0.0000**	↑	33.314	16.608–33.314	0.682	0.4712–0.8249
*IL8*	0.771	0.061	0.651–0.890	**0.0000**	↓	0.803	0.546–1.060	0.4482	0.1774–0.7190
*IL10*	0.658	0.075	0.511–0.805	**0.0354**	↑	0.485	0.302–0.536	0.4067	0.1601–0.5818
*IL13*	0.605	0.075	0.459–0.752	0.1588	ns	0.068	0.034–0.097	0.2869	0.1071–0.4021
*IL18*	0.714	0.067	0.582–0.846	**0.0014**	↑	0.327	0.031–0.353	0.4378	0.2477–0.6129
*TNFA*	0.552	0.076	0.403–0.701	0.4945	ns	8.334	0.179–8.437	0.2062	0.0357–0.4136
*RELA*	0.618	0.079	0.464–0.772	0.1328	ns	1.715	0.806–1.801	0.3134	0.0392–0.432

Significant differences (*p* < 0.05) are marked by using bold fonts. ↑ refers to a stimulant; ↓ refers to a destimulant; AUC, area under the ROC curve; 95% Cl, 95% confidence interval; ns, not significant; SE, standard error; YI, Younden index.

**Table 6 ijms-23-14677-t006:** The validity of pregnancy FPG and gene expression in predicting the AGT state at 1-year follow-up in women with prior GDM (*n* = 28).

Variable	AUC	SE	95% CI	*p*	Effect	Cut-Off Value	95% CI	YI	95% CI
FPG	0.711	0.099	0.517–0.905	**0.0328**	↑	101.0	81.0–106.0	0.333	0.100–0.456
*IL6*	0.411	0.122	0.171–0.651	0.4677	ns	264.260	1.944–264.260	0.200	0.000–0.400
*IL8*	0.750	0.112	0.53–0.970	**0.0260**	↑	0.667	0.307–0.669	0.544	0.200–0.789
*IL10*	0.650	0.107	0.441–0.859	0.1598	ns	0.618	0.089–1.343	0.278	0.056–0.400
*IL13*	0.733	0.1	0.538–0.929	**0.0193**	↑	0.492	0.105–0.553	0.444	0.167–0.611
*IL18*	0.550	0.113	0.329–0.771	0.6574	ns	1.341	0.353–2.316	0.178	0.000–0.344
*TNFA*	0.761	0.1	0.565–0.957	**0.0089**	↑	4.442	0.974–4.442	0.467	0.111–0.633
*RELA*	0.594	0.12	0.358–0.831	0.4331	ns	1.916	0.092–3.865	0.267	0.000–0.511
(*IL8* × *IL13* × *TNFA*)	0.850	0.082	0.689–1.000	**0.0000**	↑	1.353	0.089–1.353	0.644	0.300–0.844
(*IL8* × *IL13* × *TNFA* × FPG)	0.850	0.082	0.689–1.000	**0.0001**	↑	140.713	7.27–140.713	0.644	0.300–0.844

Significant differences (*p* < 0.05) are marked by using bold fonts. ↑ refers to a stimulant; AUC, area under the ROC curve; 95% Cl, 95% confidence interval; ns, not significant; SE, standard error; YI, Younden index.

**Table 7 ijms-23-14677-t007:** The parameters of the best-fitting logistic regression model for predicting postpartum AGT in GDM patients (NGT, *n* = 18; AGT, *n* = 10).

Variable	Estimate	SE	Wald’s Test	*p*	OR (95% CI)
(Intercept)	−2.230	0.759	8.636	0.0033	0.108 (0.024, 0.476)
(*IL8* × *L13* × *TNFA*)	1.422	0.579	6.037	0.0140	4.147 (1.333, 12.896)

Abbreviations: 95% CI, 95% confidence interval; OR, odds ratio; SE, standard error.

## Data Availability

The datasets used and analyzed during the current study are available from the corresponding author on reasonable request.

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
