# Peer review of "Transcriptomic Dysregulation of Inflammation-Related Genes in Leukocytes of Patients with Gestational Diabetes Mellitus (GDM) during and after Pregnancy: Identifying Potential Biomarkers Relevant to Glycemic Abnormality"

_ijms, 2022, doi:10.3390/ijms232314677_

Round 1

Reviewer 1 Report

In this paper, Zieleniak et al. assessed the gene expression levels of some cytokines in leukocytes from women with gestational diabetes mellitus or normal gestation. Moreover, the first group was analyzed also 1 year after partum. The Authors found some differences in gene expression in the leukocytes of their patients according to glucose tolerance status. Moreover, they identified a panel of genes that was associated to the development of abnormal glucose tolerance after partum.

In my opinion, the main limitation of the present paper is the absence of a post-partum control in subjects who had a normal glucose tolerance during pregnancy. Actually, many differences observed in post-partum could be due not to pathological consequences of gestational diabetes, but to the return to normality after the physiological changes occurring during pregnancy. On this issue, pregnancy has been observed to alter plasma levels of interleukin-10 (Holmes, Cytokine 2003), as well as IL-6 (Dibble, Clin Appl Thromb Hemost 2014), or IL-13 (Kanninen, Reprod Sci 2013). Similar alterations might be present in the gene expression profiles in leukocytes of these or other cytokines. Please discuss.

The Authors state that they could not detect any post-partum differences in the gene expression profiles of cytokines between women who developed abnormal glucose tolerance and women who did not. However, a gene panel was able to predict the development of abnormal glucose tolerance. Was there any difference in gene expression (and/or in other parameters) during pregnancy between women who subsequently developed abnormal glucose tolerance and women who did not?

Reviewer 2 Report

The authors presented transcriptional response of inflammation-related genes linked to metabolic phenotypes of GDM women during and after pregnancy, which has the potential a potential diagnostic classifiers for GDM and biomarkers for predicting AGT.

I think personally the paper is well written and interesting.

Major

1). I would like to know the reasones in detail to choice gene of IL-6, IL-8, 17, IL-10, IL13, IL18, TNFA and NF IL6, IL8, 17 IL10, IL13, IL18, TNFA and NFκB/RELA. Other factors such as IL-1beta, oxidative stress and advenced glycation end products(AGEs) were not measured ? If added these facotors, the manuscript would be more fulfill?

2). Glucose valiavirity between GDM and NGT is tiny in this study same as others. Do you think the number of samples is adequate or enough to investigate in this study?

Round 2

Reviewer 2 Report

The authors replied logically for my comments.

I don't have anything to discuss those.